# Hesitancy in COVID-19 Vaccine Uptake and Its Correlated Factors Using Multi-Theory Model among Adult Women: A Cross-Sectional Study in Three States of Somalia

**DOI:** 10.3390/vaccines11091489

**Published:** 2023-09-14

**Authors:** Adam A. Mohamed, Temesgen F. Bocher, Mohamed A. Magan, Cashington Siameja, Said A. Mohamoud

**Affiliations:** Save the Children International, Mogadishu P.O. Box 39664-00623, Somalia

**Keywords:** COVID-19 vaccine uptake, barriers, hesitancy, multi-theory model (MTM), Somalia

## Abstract

Background: In developing countries, access to information, awareness, and availability of COVID-19 vaccines are key challenges. Somalia launched the COVID-19 vaccination in March 2021; however, the uptake of the vaccination is slow, which creates fear of further loss of life in the country unless intentional and organized campaigning and efforts are made to improve both the availability of the vaccine and its acceptance by the community. This study aimed to understand the current level of awareness, accessibility, trust, and hesitancy toward the COVID-19 vaccine among women in Somalia. Methods: To assess COVID-19 vaccine uptake, acceptance, community awareness, and hesitancy rates in Somalia, we carried out a cross-sectional mixed methods study in three regions of Somalia that were selected randomly out of the 18 regions of Somalia. A multi-theory model (MTM) was developed to identify correlated factors associated with the hesitancy or non-hesitancy toward COVID-19 vaccination among women of all ages (18 years and above). Results: A total of 999 eligible women (333 in each district) of 18–98 years old were interviewed in March 2022. About two-thirds (63.76%) of participants reported hesitancy about receiving the COVID-19 vaccine. The theory model initiation construct indicated that behavioral confidence in the vaccine (b = 0.476, *p* < 0.001), participatory dialogue (at b = 0.136, *p* < 0.004), and changes in the physical environment (b = 0.248, *p* = 0.015) were significantly associated with COVID-19 vaccine acceptance among women who were not hesitant to take the vaccine. Conclusions: The availability of COVID-19 vaccines may not translate into their uptake. The decision to get the vaccine was determined by multiple factors, including the perceived value of the vaccination, previous experience with the vaccine, perceived risk of infection, accessibility and affordability, and trust in the vaccine itself. Public health education programming and messaging must be developed to encourage vaccine uptake among women with varying levels of vaccine hesitancy.

## 1. Background

Somalia remains one of the most complex and long-standing humanitarian crises in the world. Cycles of extreme flooding, spikes in conflict, the worst locust infestation in decades, and the outbreak of COVID-19 have all resulted in unprecedented humanitarian needs in 2020. It is estimated that the Somali population is 17 million, with 39 percent of the population living in urban areas, 23 percent living in rural areas, 24 percent in nomadic areas, and 14 percent living in internally displaced settings (IDPs) [1].

The COVID-19 pandemic has posed an unprecedented impact on the health and livelihoods of the global population, with many economies impacted, and women have been the most affected by the direct impact of the disease’s risk and the indirect effect of the pandemic that disrupted the economic situation of countries. In developing countries, access to information, awareness of the importance of the vaccine, and availability of the vaccine are key challenges. In developing countries, both the availability and uptake of the vaccine have been challenged by political, social, religious, economic, and cultural factors [2]. Hesitancy toward COVID-19 vaccination in developing countries is a less known area that requires rigorous research. Understanding the drivers to vaccine acceptance is key to identifying effective advocacy and campaigning strategies to avert the negative impact of the pandemic [3]. Recently, countries have been rolling out approved COVID-19 vaccines in a bid to contain the spread of the disease and reduce associated deaths. However, there is emerging hesitancy toward the COVID-19 vaccine that limits its acceptability and effectiveness in reducing the negative impact of the pandemic on public health. When the COVID-19 vaccine was developed, those who were at high risk of the virus where given priority of vaccination [4]. The availability of the COVID-19 vaccine may not translate into its uptake; although governments will provide the vaccine, its uptake is voluntary, and people are still hesitant about its effectiveness, side effects, and overall importance [5].

Conducting a comprehensive study to understand the supply and demand-related economic and social factors hindering the acceptance of COVID-19 vaccination among women in Somalia could highlight what are the causes of mistrust and why people are willing to accept or not accept the COVID-19 vaccine if made available. Somalia needs additional work to achieve a high and equitable uptake of vaccination by designing evidence-based behaviorally informed strategies for advocacy and campaigning. Somalia has the highest population of internally displaced people (IDP) across the Horn of Africa, with an estimated 2.6 million internally displaced people due to the insecurity, conflict, drought, and seasonal floods. Somalia lunched the first phase of COVID-19 vaccination in March 2021; however, the uptake of the vaccination is slow, which creates fear of further loss of life in the country unless intentional and organized campaigning efforts are made to improve both the availability of the vaccine and acceptance by the community.

## 2. Methods

### 2.1. Study Design and Period

A multi-theory model (MTM) was developed to identify the correlated factors associated with the hesitancy or non-hesitancy toward COVID-19 vaccination among adult women of the ages 18 years and above in three districts of Somalia (Baidoa, Galkayo, and Kismayo). The objective of this study was to generate evidence to guide effective vaccination coverage to enhance the uptake of the COVID-19 vaccination in the country. This cross-sectional study employed a multi-theory model-based approach in which negative and positive factors were intractably included to ultimately lead to the decision of getting vaccinated. This study generated evidence of the determining factors for acceptance or non-acceptance of the COVID-19 vaccine and provides recommendations on the best strategies to improving the acceptance of COVID-19 vaccination, considering the sociocultural and behavioral barriers limiting the uptake and acceptance of the COVID-19 vaccine. 

### 2.2. Study Area and Population

A community-based cross-sectional study was conducted in three randomly selected regions out of the 18 regions in Somalia. All three regions constitute large proportions of IDP influxes which have expanded due to the recent droughts and conflicts (see Appendix A). Baidoa District is the largest of the three regions and the current transitional capital of the South West State of the Federal Government of Somalia. The city is situated at the center of one of the most densely populated areas in Somalia. Galkayo District is situated in the northcentral part of Somalia and is one of the most developed towns in the region. A divided interstate city, it is sectioned along a north/south axis, with the main northern portion forming a part of the autonomous Puntland State, while the southern tip is governed by the Galmudug State of Somalia. Kismayo District is the capital of both the Lower Juba region and the Jubaland State of Somalia. The city is located on the coast of the Indian Ocean near the mouth of the Juba River approximately 500 km southwest of Mogadishu. The port city is of not only regional but also national strategic significance as it lies halfway between Mogadishu and the Kenyan border. These three selected districts exhibit typical urban settings with a mix of IDPs and hosting communities. Inclusion criteria for participation were as follows: (1) adult woman (18 years and above); (2) providing informed consent; and (3) ability to communicate in Somali.

### 2.3. Sample Size and Sampling Methods

From March to April 2022, we conducted a cross-sectional survey among women of any ages in Somalia. We randomly selected three regions (capital districts) from the 18 regions of Somalia. Within each of the three districts, we randomly selected five villages consisting of three host communities and two IDP settlements. Thus, we collected data from 15 settlements. The three districts were Baidoa in the South West State, Kismayo in Jubaland State, and North Galkayo in the autonomous state of Puntland. A map of the selected districts, villages chosen to collect data, and sampling distributions can be accessed in Appendix A. The sample size was allocated equally for each of the three districts. We targeted a sample size of 385 per district using [6] a formula that initially gave us a total of 1155 households. We targeted more households to be included in the data collection than the required sample size using a 95% confidence interval, with a 50% proportion as likely to be vaccinated since we did not have a prior study proportion in Somalia. 

The study applied a systematic sampling technique to select the households with adult woman (18 and above). The first household was randomly chosen based on its location in the approximate geographical center of the IDPs or villages. The enumerators proceeded to the next available households nearby that were mapped in the villages or IDPs. We had a single respondent selection per household and for households with more than one eligible participant, we randomly selected one woman among those present at the time of the visit. Eligible women per household are described as those who had lived for at least one month under the same roof and shared cooking and eating facilities from the same source.

### 2.4. Ethics

Ethical approval was obtained from the Ministry of Health and Human Services (Reference number: MOH&HS/DGO/0321/March/2022). All participants provided informed consent before their participation after it was explained to them that their participation was voluntary and that the information obtained would only be used for the purpose of this research. All efforts were made to ensure the confidentiality of their responses.

### 2.5. Instrument Development and Measures

This study adopted open-ended questionnaire guides that were designed around COVID-19 vaccine hesitancy and uptake [7]. The survey instrument consisted of sociodemographic items (e.g., age, education, wealth, residence, etc.), vaccine hesitancy, mistrust, and willingness to take the vaccine (i.e., COVID-19 vaccine trust, belief in the effectiveness of the vaccine, decision to take the vaccine, etc.). To ensure the reliability of the study tool, we engaged a timely adjustment and review of the tools, and all needed proper amendments were made to make sure that the tool elicited precise answers to the right questions. The survey tool was tested on a group of women for readability and it was found to be easily understood. 

### 2.6. Outcome Measurement and Explanatory Variables

The main dependent variable of the study was willingness to get the vaccine if made available. Respondents were asked if they were willing to receive the COVID-19 vaccination for themselves. The expected response for the willingness question was ‘no intention to receive vaccination’, ‘undecided’, or ‘intention to receive the vaccination’ see Table 1 below. The intent to get vaccinated was considered as ‘vaccine acceptance’ whereas uncertainty and unwillingness to get vaccinated were considered as ‘vaccine hesitancy’ and ‘vaccine unacceptance’, respectively. The study also captured the respondents’ awareness of the availability of vaccination, knowledge and attitude toward the vaccination, and previous vaccination experience.

## 3. Results

### 3.1. Sociodemographic Characteristics of Respondents

Cross-sectional surveys were conducted in three districts of Somalia (Baidoa, Galkayo, and Kismayo). A total of 999 eligible women (333 in each district) of the ages between 18–98 years were interviewed in March 2022. The study used a validated and reliable 56-item questionnaire exploring the respondents’ demographics, experience with COVID-19 disease, awareness about the perceived advantages and disadvantages of taking the COVID-19 vaccination and initiation to take the vaccine if made available see Table 2 below. It was found that 31% of the surveyed women were from IDPs and 69% from host communities or urban settlements as you can see from Table 3. The average age of the surveyed women was 41 years, the youngest being 18 years, and the oldest being 98 years. A total of 63% of the women was married, 15% widowed, 13% divorced, and 9% single. Meanwhile, 55% did not have any kind of education, 21% had completed some kind of informal school, 16% completed primary school, and only 7% completed high school and above. About eight in 10 (82%) considered themselves as unemployed, 12% as employed in a self-owned small business, and 5% as employed in a formal paid job. On average, the surveyed mothers reported having five children and 35% of the surveyed women were the head of the household (30% in Baidoa and Kismayo and 46% in Galkayo). About one-third of the respondents had a monthly family income below USD 50, 36% between USD 50–150, 25% between USD 150–250, and less than 5% had a monthly income of USD 250 and above. 

### 3.2. Results of the MTM Model Analysis

The results of the multi-theory model initiation construct indicate that behavioral confidence in the vaccine (b = 0.476, *p* < 0.001), participatory dialogue (i.e., the difference between the perceived advantages and disadvantages of taking the COVID-19 vaccine, at b = 0.136, *p* < 0.004), and changes in the physical environment (i.e., access, affordability, and willingness to take the vaccine, b = 0.248, *p* = 0.015) were significantly and positively associated with COVID-19 vaccine acceptance among those women who were not hesitant to take the vaccine. Moreover, three attributes (participatory dialogue, physical environment, and behavioral confidence) accounted for 47% of the variation among the women who were non-hesitant to the vaccine regarding the acceptance of COVID-19 vaccination. On the contrary, participatory dialogue played an insignificant role (b = 0. 0267, *p* < 0.001) for those who were hesitant toward COVID-19 vaccination; thus, the decision not to take the vaccine is less dependent on the compared advantages and disadvantages but rather, the lack of confidence in taking the vaccine (in addition to its side effects, worries about a lack of long-term studies, b = 0.478, *p* = 0.001) and changes in physical environment (improved access, affordability, and willingness to take the vaccine, b = 0.256, *p* = 0.001) played key roles in building a negative attitude toward the vaccine. The three attributes account for 56% of the variation among the women who were hesitant to accept COVID-19 vaccination.

The multi-theory model would be best utilized to enhance the uptake of COVID-19 vaccination intervention including in the design of messaging. We examined factors explaining the hesitancy and non-hesitancy among women by posing questions assessing their opinion on their willingness to get the vaccine if the vaccine was made available, the perceived advantages and disadvantages, changes in the current status of the vaccine’s availability, the accessibility of the vaccine, experience with previous vaccination, and on what would play a positive role in improving their acceptance of the vaccine. The analysis conducted to examine the differences in the constructs of the MTM for non-hesitant and hesitant respondents indicated that initiation toward vaccination was higher among the non-hesitant individuals; likewise, the mean values for multiple theory measures were significantly higher among individuals exhibiting no hesitancy in vaccine acceptance. The measure of pairwise correlation between the constructs of the MTM indicated that the initiation to receive COVID-19 vaccination was significantly influenced by behavioral confidence in the vaccine (r = 0.721, *p* < 0.001) and changes in the physical environment (r = 0. 0.679, *p* < 0.001) for hesitant individuals. 

Thus, building behavioral confidence in the side effects and existing evidence on the effectiveness of the vaccine, and improving the availability and accessibility of the vaccine to improve overall confidence and hence lead to a higher uptake of the vaccination are of paramount importance. Similarly, among non-hesitant individuals, behavioral confidence (r = 0.483, *p* < 0.001) and changes in the physical environment (r = 0.480, *p* < 0.001) were significantly associated with the initiation construct. This study provides evidence for the utilization of a multi-theory model (MTM) as part of evidence-based campaigning and advocacy work to reach the most vulnerable women in Somalia and provide them with the COVID-19 vaccination. In all three districts, at least 30% of the respondents were from IDP centers. The unemployment rate in the study area was very high among women, with 82% of the survey respondents being unemployed and only about 15%, 12%, and 10%, in Galkayo, Baidoa, and Kismayo, respectively, being employed in their own business. 

From observing the non-vaccine-related factors influencing the decision to get COVID-19 vaccination or not, the poorest women and those living in IDPs were more likely to intend to get the COVID-19 vaccine compared to the other income groups. Furthermore, those who already knew someone who had gotten vaccinated and heard about the existence of the vaccination were more likely to intend to receive the vaccination. On the other hand, widows, residents of Baidoa District, lactating women, and pregnant mothers were more likely to show no intention to receive the vaccine. Estimates of the group Likert scale differences attained significance after conducting t-tests. The overall reliability of the constructs was a 0.56 scale reliability coefficient indicating that the variables were significantly different. To further validate and examine the findings of the descriptive analysis and MTM model, a multi-regression model were conducted. A bivariate probit model was developed to measure the determinants of the likelihood of trust in the COVID-19 vaccine and a multinomial logit model was used to assess hesitancy levels (i.e., intention to get vaccinated, no intention, or undecided). 

### 3.3. Bivariate Analysis of the Determinants of Trust toward the COVID-19 Vaccine

Trust in the COVID-19 vaccine and vaccine hesitancy were explained based on demographics (age, marital status, current health situation, COVID-19 awareness, and information about the COVID-19 pandemic). Therefore, efforts to vaccinate higher risk older adults must aim not only to educate and provide vaccine access but boost trust in the vaccine development process and vaccine effectiveness. Those who had contracted COVID-19 and obtained a positive result were 15% more likely to trust the COVID-19 vaccine, compared to those who did not contract the virus. Experience with positive COVID-19 led to 13% more trust in the COVID-19 vaccine considering variables included in the analysis. Experience with previous vaccination for any disease had a positive impact on building trust and reducing vaccine hesitancy. 

The study results indicate that previous vaccine experience was associated with a 21% higher probability of trusting in the Covid-19 vaccine. Those respondents who had a close family member or friend who had contracted COVID-19 were 5% more likely to trust the COVID-19 vaccine, compared to those who did not have family member infected with COVID-19. The probit analysis showed a strong positive correlation between exposure to the vaccine and trust in the vaccine. Those who knew someone vaccinated were 57% (CI: 0.38–0.75, *p* < 0.015) more likely to trust in the COVID-19 vaccine. Having information about the COVID-19 vaccination was significantly associated with trust in the vaccine and prior information about the COVID-19 vaccination led to a 16% greater chance to trust in the vaccination. In the study areas, 18% of respondents indicated having received at least one dose of the COVID-19 vaccine; receiving vaccination was also strongly associated with trust in the vaccine. 

This study suggests that trust around COVID-19 is highly influenced by place of residence, marital status, receiving at least one dose of the COVID-19 vaccine, previous vaccine experience, and knowledge about the COVID-19 virus, and exposure to COVID-19 vaccine information in women. Trust levels varied significantly between the women living in host communities or urban settlements and the women living in internally displaced people settings (IDP were 31% more likely to trust the COVID-19 vaccine compared to women in urban or host communities) see Table 4 below. Residents of Galkayo District were 36% more likely to trust the vaccine and Kismayo residents were 30% less likely to trust the COVID-19 vaccine compared to their Baidoa counterparts. Education level played a positive role in the decision to receive the vaccine and women without education did not intend to receive the COVID-19 vaccine. Mistrust was prevalent among the host communities, Kismayo residents, widowed, and those who had little to no information on COVID-19 or its vaccine. 

Those who already received a vaccine for any disease in the past were 24% less likely to hesitate to take the COVID-19 vaccine, IDPs were 30% less likely to receive the vaccination compared to the host communities, knowing someone who had gotten the COVID-19 vaccination was 40% associated with enhanced confidence in the vaccination, and those who had heard of the existence of the vaccination were 32% less likely to be hesitant toward the vaccination. Thus, using public figures who have received vaccination and improving access to information on the existence of the vaccine and its benefits play key roles in reducing hesitancy and improving confidence among the community. A total of 82% of respondents had not received COVID-19 vaccination at the time the survey was conducted. For some, hesitancy around vaccination was grounded in insufficient knowledge, lack of confidence in the benefits of vaccination, or overconfidence in one’s ability to avoid the disease in question. Scores for behavioral confidence, changes in physical situation (availability), initiation intention, and advantages and disadvantages enable factoring out the decision-influencing factors.

## 4. Discussion 

This study aimed at understanding the current level of awareness, accessibility, trust, and hesitancy toward the COVID-19 vaccine in Somalia. This study was conducted to evaluate the level of acceptance of the COVID-19 vaccine and its determinants among women of all ages (18–98). The findings of the study will inform governments and COVID-19 response actors to design appropriate and evidence-based vaccination coverage expanding current strategies and policies. Our study identified similar results underpinning COVID-19 vaccine hesitancy and mistrust compared to a study conducted in Ethiopia [8]. The decision to get the vaccine is influenced by multiple factors including the perceived value of the vaccination, previous experience with the vaccine, the perceived risk of the infection, accessibility and affordability, and trust in the vaccine itself [9]. Unless appropriate and expedited actions are taken to increase the uptake of the vaccine and improve its availability, the risk of the pandemic spread will overwhelm the health systems’ abilities.

In our study, 63.7% of study participants were hesitant to take the vaccine. A similar vaccine hesitancy rate was reported by a study conducted in Nigeria [10]. Over thirty-five percent (36%) of the participants reported their willingness and intention to take the vaccine, which is consistent with the findings from other studies conducted in other countries [11]. From observing the non-vaccine-related factors in influencing the decision to take the COVID-19 vaccine or not indicated that the poorest women and those living in IDPs were more likely to intend to get the COVID-19 vaccine compared to other income groups; those who already knew someone who had gotten vaccinated and heard about the existence of the vaccine intended to receive the vaccination. The analysis conducted to examine the differences in constructs of the MTM for non-hesitant and hesitant respondents indicated that the initiation of vaccination was higher among the non-hesitant individuals; likewise, the mean values for multiple theory measures were significantly higher among individuals exhibiting no hesitancy toward vaccine acceptance. This was a similar result to a study conducted in America [12]. 

In a study conducted in Somalia focusing on the IDP population, people with higher earnings in the previous week were more likely to say they had difficulty paying rent due to COVID-19 [13].

A recent study conducted among 15 countries in Sub-Saharan African countries indicated that the acceptance of the COVID-19 vaccine varies from country to country. For instance, Ethiopia and Niger had high acceptance rates (94% and 93%, respectively) while Senegal and the Democratic Republic of Congo had an acceptance rate of 65% and 59%, respectively [14]. Compared to these findings, our study found that only 36% of the interviewed women would accept the COVID-19 vaccine if made available and accessible. A solely cross-sectional study among nine low- and middle-income countries showed that the prevalence of COVID-19 vaccine acceptance varied from the lowest being 76.4% to the highest being 88.8% [15]. On the contrary, our study is generally in line with a study published in 2021 in Italy of elderly people, where the vaccine acceptance frequency and rate was found to be 460 (97.9) [16].

A previous study conducted in Somalia reported a relatively good knowledge of COVID-19 by IDP communities. Our study is relatively correspondent to this finding [17]. This is likely due to radio campaigns and the distribution of translated posters by local governments and NGOs, aimed at improving awareness of COVID-19 and its prevention [13]. The pandemic is associated with a high prevalence of stress, depression, and anxiety which can be greater in precarious low-resource settings [18]. The recent Somalia Health and Demographic Survey (SHDS 2020) reported that a lack of sanitation and handwashing facilities is a major risk for coronavirus transmission [19]. Our study found that participants were sometimes misinformed about disease symptoms by health workers or community committees, but more importantly, some lacked trust in authorities, healthcare services, and humanitarian responders. This was the same conclusion of a previous study conducted in Somalia [20].

## 5. Conclusions and Recommendations

The availability of COVID-19 vaccines may not translate into its uptake. The decision to get the vaccine was determined by multiple factors, including the perceived value of the vaccination, previous experience with the vaccine, perceived risk of infection, accessibility and affordability, and trust in the vaccine itself. Developing public health education programming and messaging to encourage vaccine uptake among women with varying levels of vaccine hesitancy is essential. Educational interventions should be designed toward women and children, promoting the COVID-19 vaccine and other vaccine acceptance. Evidence-based interventions should be designed and disseminated to promote COVID-19 vaccine and other vaccine acceptability for all mothers. Based on our study findings, it can be concluded that the multi-theory model (MTM) can be an effective tool for developing public health education programming and messaging to encourage vaccine uptake among mothers and their children, with varying levels of vaccine hesitancy. For instance, working on behavioral change communication and social mobilization toward COVID-19 vaccines using culturally appropriate interventions might improve attitudes and perceptions, which in turn would increase the rate of COVID-19 vaccination and other childhood vaccinations. Furthermore, to increase positive social norm perception, people who influence behavior such as community leaders and health workers should be engaged and put at the forefront of campaigns. 

## Figures and Tables

**Table 1 vaccines-11-01489-t001:** Hesitancy, mistrust, and willingness toward COVID-19 vaccine uptake.

Characteristic	Frequency	Percentage
Access to COVID-19 awareness information		
No or low access	346	35%
Moderate access	475	47%
High access	178	18%
Willingness to take the COVID-19 vaccine		
No intention	116	12%
Undecided	521	52%
Intention	362	36%
Trust toward the new COVID-19 vaccine		
Not or a little trusted	455	46%
Moderately trusted	375	37%
Very trusted	169	17%
Importance of the COVID-19 vaccine to your health		
Not at all important	120	12%
A little important	394	39%
Moderately important	485	49%
Do you think the benefits of the COVID-19 vaccine outweigh the side effects?		
Yes, I do think	627	63%
No, I don’t think	372	37%
COVID-19 vaccine is not safe and is a risk to your health		
Not or hardly safe	391	40%
Moderately safe	324	32%
Very safe	284	28%

**Table 2 vaccines-11-01489-t002:** Access to information about and awareness of the COVID-19 vaccine.

Characteristic	Frequency	Percentage
Do you have any information about the COVID-19 vaccine?		
Yes, I have heard about the new vaccine	762	76%
No, I don’t have any information	237	24%
How easy is it to get a COVID-19 vaccination in your locality?		
Not or hardly easy	395	39%
Moderately easy	455	45%
Very easy	149	14%
Do you have access to the COVID-19 vaccine in your locality?		
Yes, I can access the COVID-19 vaccine	736	74%
No, I can’t access the COVID-19 vaccine	263	26%
If you can’t access the vaccine, what are the barriers (Total = 263)?		
Not available	145	55%
Far away from my locality	104	40%
Other reason	14	5%
Have you received the COVID-19 vaccine?		
Yes	180	18%
No	819	82%
Do you know where to get the COVID-19 vaccine?		
Yes, I know	543	66%
No, I don’t know	276	34%

**Table 3 vaccines-11-01489-t003:** Sociodemographic characteristics and profiles of respondents.

Characteristic	Baidao	Galkayo	Kismayo	Total
Residence				
Host or urban setting	232	231	230	693
IDPs	101	102	103	306
Employment				
Group business	4	0	6	10
Paid employment	19	17	9	45
Self-employment	39	49	34	122
Unemployed	271	267	284	822
Marital status				
Divorced	36	47	43	126
Married	220	186	226	632
Single	25	44	22	91
Widowed	52	56	42	150
Pregnancy status				
Elderly	61	103	136	300
Lactating	109	73	87	269
Pregnant	55	47	38	140
Other	107	110	71	288
Age				
18–30	113	131	80	324
31–45	130	101	116	347
46–60	54	70	70	194
61 and above	36	31	67	134
Education				
College	1	12	1	14
Informal education	84	76	45	205
No education	190	135	233	558
Primary	38	82	38	158
Secondary	20	28	16	64
Family income per month (USD)				
Less than 50	141	39	113	293
50–150	132	112	113	357
150–250	59	100	96	255
250–350	1	45	11	57
Above 350	0	37	0	37
Have you been tested for COVID-19?				
Yes	65	29	32	126
No	268	304	301	873
Have any of your family or friends had COVID-19?				
Yes	146	103	55	304
No	187	230	278	695
Do you have any chronic diseases?				
Yes	56	53	38	147
No	277	280	295	852

**Table 4 vaccines-11-01489-t004:** Determinants of trust in the COVID-19 vaccine.

Dependent = Trust in COVID-19 Vaccine (Yes = 1 No = 0)				
Characteristic	Coef.	Std. Err.	z	*p* > z
Place of residence (Reference = Host community)				
IDPs	0.313	0.097	3.220	0.001
District (Reference = Baidoa)				
Galkayo	0.361	0.110	3.280	0.001
Kismayo	−0.295	0.112	−2.630	0.008
Age of the respondent in years	0.002	0.003	0.670	0.502
Marital status (Reference = divorced)				
Married	0.006	0.132	0.050	0.961
Single	−0.146	0.192	−0.760	0.449
Widowed	−0.291	0.175	−1.660	0.097
Received COVID-19 vaccine?				
Yes	0.904	0.138	6.550	0.001
Previous vaccine experience				
Yes	0.212	0.091	2.330	0.020
Tested COVID-19 positive				
Yes	0.132	0.139	0.950	0.343
Know anyone infected with COVID-19 virus				
Yes	0.246	0.101	2.430	0.015
Know anyone vaccinated against COVID-19				
Yes	0.566	0.092	6.160	0.001
_cons	−0.533	0.221	−2.410	0.016

## Data Availability

All the data related to this study can be made available from the corresponding author (Adam A. Mohamed). See also the data availability policy from “MDPI Research Data Policies” at https://www.mdpi.com/ethics (accessed on 15 March 2023).

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
