# Peer review of "Hesitancy in COVID-19 Vaccine Uptake and Its Correlated Factors Using Multi-Theory Model among Adult Women: A Cross-Sectional Study in Three States of Somalia"

_vaccines, 2023, doi:10.3390/vaccines11091489_

Round 1

Reviewer 1 Report

Dear Authors, 

thank you for your work. It investigates a topic of maximum interest, as vaccination needs to be boosted by a better understanding of hesitancy mechanisms. 

The strengths of your work regard the practical implications of your findings. The flaws are the following:

- The abstract seems a bit too long and does not need to report the numerical results of analyses. Please consider to reduce it; 

- I would integrate the footnote of page 2 in the main text; 

- Line 189: please fix the reference; 

- Line 215: please replace "covid19" with "COVID-19"; 

- All the comments of results should be erased from the Results section and moved to the Discussion section; 

- A Conclusion paragraph should be added. 

I suggest publication after minor revision.

Author Response

Reviewer 1 comments were addressed.

thank you for your work. It investigates a topic of maximum interest, as vaccination needs to be boosted by a better understanding of hesitancy mechanisms. 

The strengths of your work regard the practical implications of your findings. The flaws are the following:

- The abstract seems a bit too long and does not need to report the numerical results of analyses. Please consider reducing it. 

Answer: Adjusted and reduced

- I would integrate the footnote of page 2 in the main text. 

Answer: integrated and removed from the footnote

- Line 189: please fix the reference. 

Answer: Fixed

- Line 215: please replace "covid19" with "COVID-19”. 

Answer: Replaced

- All the comments of results should be erased from the Results section and moved to the Discussion section; 

Answer: Done

- A Conclusion paragraph should be added. 

Answer: Added

I suggest publication after minor revision.

Reviewer 2 Report

Estimated Authors of the paper "Hesitancy in COVID-19 Vaccine uptake and its correlated factors using Multi-theory Model among adult women: A cross-sectional study in three states of Somalia". I've read your paper with great interest for a series of reasons. More precisely, the settings, the background theory (i.e. MTM, https://www.researchgate.net/publication/282155951_Multi-theory_model_MTM_for_health_behavior_change), are quite unusual when dealing with SARS-CoV-2 vaccination, fully deserving therefore an extensive appraisal.

Still, the present paper, in its current stage of development, is affected by substantial shortcomings that collectively force this reviewer in urging a further stage of revision.

More precisely:

1) please explain how the questionnaire was actually delivered: were the investigators involved? was the questionnaire delivered to the potential participants able to read and write and then retrieved? etc.

2) Table 1 should implement crude numbers of participants by geographic areas, and further statistical analyses are required as well: please include a chi square test analysis for distribution of the variables by geographic areas, in order to explain whether eventual results were likely associated with some sort of demographic and socio-economic features of the participants. This is of particular significance as the Authors have then identified the geographic area among the main effectors of vaccine hesitancy.

3) Similar improvements are required for data included in  Table 2 and 3

4) a p value = 0.000 does not exist; this is a quite common mistake from the data reporting of some statistical software (e.g. SPSS) that instead of a more appropriate p < 0.001 prefer the aforementioned and incorrect reporting, that should be fixed across the text; please note that "b=0. 0267, p<0.283" does not make any sense; please double check.

5) neither domains included in the analyses nor the statistical tests themselves were preventively explained, the reader therefore may be quite confused by the results section: a more appropriate approach requires that Authors expand in full details how the domains of their analyses were calculated, and how the data were handled, including the actual statistical analyses they've performed.

6) Table 4 is unclear, and the caption should be reframed in order to provide the reader some more accurate explanation making the table indipendent from the main text.

7) Discussion should be reframed by assessing individuals steps of the achieved results, and then sharing all possible comments on the available literature and similar studies either performed in such a settings or through the very same or similar instruments; to date, discussion is too vague and does not add any substantial value to the main text.

8) please include some maps across the main text in order to explain where the study was performed within Somalia

9) Authors should provide a flow-chart including the rate of participation of invited individuals since the inception of the study until the eventual definition of the study sample. In other words: how many women were initially recruited? how many of them eventually participated into all the stages of the questionnaire? Even the participation rate has a clear significance when dealing with studies on vaccine hesitancy.

Author Response

  • please explain how the questionnaire was actually delivered: were the investigators involved? was the questionnaire delivered to the potential participants able to read and write and then retrieved? etc.

Answer: The study investigators trained a data collection team consisting of both males and females. To ensure the reliability of the study tool, we engaged a timely adjustment and review of the tools. All needed proper amendments were made to ensure that the tool was eliciting a precise answer to the right questions. The survey tool was tested with a group of women for readability, and it was found that comprehension was easily understood.

  • Table 1 should implement crude numbers of participants by geographic areas, and further statistical analyses are required as well: please include a chi square test analysis for distribution of the variables by geographic areas, in order to explain whether eventual results were likely associated with some sort of demographic and socio-economic features of the participants. This is of particular significance as the Authors have then identified the geographic area among the main effectors of vaccine hesitancy.

Answer: Table 1 is showing the number of participants per location or per state. We used the percentage for a summary which is more reader understanding and friendly.

  • Similar improvements are required for data included in Table 2 and 3

Answer: Addressed

  • a p-value = 0.000 does not exist; this is a quite common mistake from the data reporting of some statistical software (e.g. SPSS) that, instead of a more appropriate p < 0.001 prefer those mentioned above and incorrect reporting, that should be fixed across the text; please note that "b=0. 0267, p<0.283" does not make any sense; please double check.

Answer: Checked and addressed

  • neither domains included in the analyses nor the statistical tests themselves were preventively explained, the reader therefore may be quite confused by the results section: a more appropriate approach requires that Authors expand in full details how the domains of their analyses were calculated, and how the data were handled, including the actual statistical analyses they've performed.

Answer: addressed and improved

  • Table 4 is unclear, and the caption should be reframed in order to provide the reader some more accurate explanation making the table independent from the main text.

Answer: Checked and addressed. The reviewer should see also the supplementary documents in this study that are attached with.

  • Discussion should be reframed by assessing individuals steps of the achieved results, and then sharing all possible comments on the available literature and similar studies either performed in such a settings or through the very same or similar instruments; to date, discussion is too vague and does not add any substantial value to the main text.

Answer: Checked and Reframed please see.

8) please include some maps across the main text in order to explain where the study was performed within Somalia

9) Authors should provide a flow-chart including the rate of participation of invited individuals since the inception of the study until the eventual definition of the study sample. In other words: how many women were initially recruited? how many of them eventually participated into all the stages of the questionnaire? Even the participation rate has a clear significance when dealing with studies on vaccine hesitancy.

Answer: Kindly see the supplementary document attached to the manuscript that is clearly indicating the response rate.

The following formula was used to calculate the response rate.

Where, RR1 = Response rate

       I = Complete interview 

      P = Partial interview

      R = Refusal and break-off

   NC = non-contact

     O = Other

  UH = Unknown if household/occupied HU

               UO = Unknown, other

The details of these quantities for our study are given in the following:

I and P=The in-person household survey was conducted in which housing units are sampled from an address-based sampling frame of 15 selected villages using systematic sampling technique. We consider less than 50% of all applicable questions answered (with other than a refusal or no answer) equals break-off, 50%-80% equals partial, and more that 80% equals complete. We found complete answered from 999 participants (i.e., I=999). We found 81 of the participants did not complete the questionnaire (i.e., P=81). 

R= Refusals and breakoffs consist of cases in which some contact has been made with the housing unit and a responsible household member has declined to do the interview, or an initiated interview results in a terminal break-off (i.e., R=47).

NC= non-contacts in in-person household surveys consist of three types: a) unable to gain access to the building, b) no one reached at housing unit, and c) respondent away or unavailable (i.e., NC=9).

O= Other cases represent instances in which the respondent is/was eligible and did not refuse the interview, but no interview is obtainable because of a) the respondent is physically and/or mentally unable to do an interview; b) miscellaneous other reasons. We did not face any language problem to exclude participants. (i.e., O=12).

UH= Cases of unknown eligibility and no interview include situations in which it is not known if an eligible housing unit exists and those in which a housing unit exists (i.e., UH=1).

UO= Not eligible cases for in-person household surveys include a) out-of-sample housing units; b) not-a-housing unit; c) vacant housing units; d) housing units with no eligible respondent; and e) situations in which quotas have been filled.  In a systematic sampling technique, we found a total of 6 households without any adult respondents during the interview (UO=6).

Thus, the response rate is, RR=999/ (999+81+47+9+12+1+6) = 86.5%

Round 2

Reviewer 2 Report

Estimated Authors, 

Even though I've appreciated the efforts to cope with my recommendations, please note the following shortcomings from the revised version of your otherwise very interesting paper:

1) in my previous review I did suggest the implementation of "some maps across the main text in order to explain where the study was performed within Somalia" and "Authors should provide a flow-chart including the rate of participation of invited individuals since the inception of the study until the eventual definition of the study sample. In other words: how many women were initially recruited? how many of them eventually participated into all the stages of the questionnaire? Even the participation rate has a clear significance when dealing with studies on vaccine hesitancy".

Your answer was: "Kindly see the supplementary document attached to the manuscript that is clearly indicating the response rate" 

Please note that you did not answer my concerns: not only maps were not implemented, but the response rate has nothing to do with the flow chart I've recommended the implementation of.

Please address this topics;

2) In my previous review I've specifically addressed Table 4, as it seemed quite confused. In your reply you suggested that "The reviewer should see also the supplementary documents in this study that are attached with". Please understand that, even through the mirror of the supplementary documents, table 4 remains unclear. Please include some of the explanations contained in the supplementary materials either as annex material (i.e. text that will be included in the final article as an appendix) of across the main text of the caption of the main table.

3) I renew my doubts about Table 1: according to STROBE statement, not only percent values but also crude values must be reported. Therefore, I urge for reformatting Table 1 accordingly, as you have done accurately and well properly for Table 2-3.

4) the explanation about the delivery of the questionnaire is appreciable, and should be included in the main text (i.e. The study investigators trained a data collection team consisting of both males and females. To ensure the reliability of the study tool, we engaged a timely adjustment and review of the tools. All needed proper amendments were made to ensure that the tool was eliciting a precise answer to the right questions. The survey tool was tested with a group of women for readability, and it was found that comprehension was easily understood.)

5) Please understand that when a certain explanation or a certain information is provided by means of supplementary material, it should be properly referenced across the text.

6) references should be extended to some similar studies, that should be implemented in the discussion and discussed accordingly. For example:  

https://pubmed.ncbi.nlm.nih.gov/36767930/

https://pubmed.ncbi.nlm.nih.gov/36011464/

https://pubmed.ncbi.nlm.nih.gov/35475006/

https://pubmed.ncbi.nlm.nih.gov/36530665/

https://pubmed.ncbi.nlm.nih.gov/36231447/

https://pubmed.ncbi.nlm.nih.gov/36177216/

https://pubmed.ncbi.nlm.nih.gov/35850783/

https://pubmed.ncbi.nlm.nih.gov/34064159/

Author Response

Reviewer 2 Comments were addressed.

Estimated Authors of the paper "Hesitancy in COVID-19 Vaccine uptake and its correlated factors using Multi-theory Model among adult women: A cross-sectional study in three states of Somalia". I've read your paper with great interest for a series of reasons. More precisely, the settings, the background theory (i.e. MTM, https://www.researchgate.net/publication/282155951_Multi-theory_model_MTM_for_health_behavior_change), are quite unusual when dealing with SARS-CoV-2 vaccination, fully deserving therefore an extensive appraisal.

  • in my previous review I did suggest the implementation of "some maps across the main text in order to explain where the study was performed within Somalia" and "Authors should provide a flow-chart including the rate of participation of invited individuals since the inception of the study until the eventual definition of the study sample. In other words: how many women were initially recruited? how many of them eventually participated into all the stages of the questionnaire? Even the participation rate has a clear significance when dealing with studies on vaccine hesitancy".

Answer: I have added map on study locations in Somalia

  • I renew my doubts about Table 1: according to STROBE statement, not only percent values but also crude values must be reported. Therefore, I urge for reformatting Table 1 accordingly, as you have done accurately and well properly for Table 2-3.

Answer: I have changed the percentages into real numbers as per your advice in table 1.

  • In my previous review I've specifically addressed Table 4, as it seemed quite confused. In your reply you suggested that "The reviewer should see also the supplementary documents in this study that are attached with". Please understand that, even though the mirror of the supplementary documents, table 4 remains unclear. Please include some of the explanations contained in the supplementary materials as annex material (i.e. text that will be included in the final article as an appendix) of across the main text of the caption of the main table.

Answer: Checked and addressed. I have removed table 4 from the manuscript.

  • Discussion should be reframed by assessing individuals steps of the achieved results, and then sharing all possible comments on the available literature and similar studies either performed in such a settings or through the very same or similar instruments; to date, discussion is too vague and does not add any substantial value to the main text.

Answer: Checked and Reframed please see.

Round 3

Reviewer 2 Report

Estimated Authors,

Most of my concerns were either addressed or solved by the present version of your paper.

No more improvements are required and I'm rather happy to endorse the eventual acceptance of this interesting contribution.

Author Response

Thank you for your comments.